# S0 Galaxies: Outer Gas Accretion through Tidal Interaction and Minor Merging

Olga Sil'chenko [1,*] , Alexei V. Moiseev [1,2] , Alexandrina Smirnova [2] and Roman Uklein [2]

1 Sternberg Astronomical Institute, Lomonosov Moscow State University, University Av. 13, 119991 Moscow, Russia; moisav@gmail.com
2 Special Astrophysical Observatory, 369167 Nizhnij Arkhyz, Russia; ssmirnova@gmail.com (A.S.); uklein@sao.ru (R.U.)
* Correspondence: olga@sai.msu.su

**Abstract:** To clarify the sources of outer gas accretion onto disk galaxies, we study the vicinity of four interacting galaxy systems in the Hα emission line by using the scanning Fabry–Perot interferometer of the 6m telescope of the Special Astrophysical Observatory RAS. We find perspective accretion flows seen as ionized-gas emission filaments between the galaxies. We discuss the whole kinematics and origin of these flows.

**Keywords:** disk galaxies; galactic structure; star formation; galaxies: kinematics and dynamics

## 1. Introduction

Today, the paradigm that disk galaxy evolution is governed by permanent gas accretion from outside is commonly accepted [1–3]. Neither the chemical evolution of galactic disks [4–6], nor the star formation histories of nearby spiral galaxies [7–9] can be explained without suggestion that all these processes are fed by outer cold gas inflow. But sources of this accretion are still disputable. There are three main possible scenarios: tidal gas exchange with neighboring galaxies, including satellite acquisition (multiple minor merging), fueled by cosmological gas–dark matter filaments which are suggested to be a part of a large-scale structure of the Universe [10,11], and so-called "hot accretion" resulting from the cooling of gas of the extended galactic halos [12–14]. As for the former scenario, which is especially attractive for astronomers because it can be observed directly and tested in astronomical observations, it has been claimed to be an insufficient source by Sancisi et al. [15], after a deep search in the 21 cm line for neutral hydrogen in the vicinity of nearby spiral galaxies. However, further persistent studies of wide areas around interacting galaxies over the full spectral range remain necessary to correctly estimate the potential of this gas accretion source.

Interacting galaxies as a class were discovered by Vorontsov-Velyaminov; in 1959, he published his first catalog and atlas of interacting galaxies after his careful inspection of the all-sky Palomar survey maps [16]. Soon, it became clear that galaxy interactions play important roles in the galaxy's evolution. In particular, the gas exchange during tidal encounters and minor merging can provide the supply of outer cold gas. The first simulations of galaxy gravitational interaction by Toomre and Toomre [17] demonstrated a rich collection of tails and bridges formed even without gas participation in the interactions. Later, a significant amount of other simulations were made, both pure N-body ones (e.g., [18,19]) and those taking into account gas dynamical effects in gas-rich galactic disks (e.g., [20–22]), and outer tails and bridges were obtained in these simulations providing gas exchange between interacting galaxies. The observations of some spectacular interacting galaxy systems also demonstrated extended outer gaseous structures, connecting the interacting galaxies—sometimes those formed from cold neutral hydrogen and seen only in the 21 cm line without optical counterparts [23–25], and sometimes star formation was detected in

these structures beyond the galactic disks, and they were excited by young stars and seen both in UV [26] and in the ionized-gas optical emission lines (e.g., [27–30]).

We continue to study outer gaseous structures in interacting galaxy systems which may be considered as possible manifestations of gas exchange between galaxies of different masses and morphological types. We use 3D spectroscopy to map emission-line intensities and line-of-sight velocities that allow us to inspect warm ionized gas distributions and to detect gas flows over large fields of view inhabited by interacting galaxies.

## 2. The Sample

For this consideration, we have chosen four interacting galaxy systems at different stages of interaction, of different stellar masses and in different types of environment. Such a small sample cannot be full or representative but the variety of the galaxy characteristics, together with their common property—the presence of a very close neighbor—makes it possible to delineate a range of possibilities of gas cross-fueling that is important for the general scenario of outer gas supply for disk galaxies. The full-colored images of the objects of our sample with their closest neighbors are presented in Figure 1, and their global characteristics taken from popular databases are given in Table 1.

In our sample, we have two giant early-type galaxies, NGC 3921 and UGC 8671, one moderate-luminosity early-type spiral UGC 1020 (though it looks like a lenticular), and bona-fide S0-dwarf UGC 7020A. NGC 3921 (Arp 224), being itself an obvious merger, is also a center of a galaxy group, with some rather massive close neighbors and demonstrating no morphological signs of interaction with NGC 3921. UGC 8671 has a merging galaxy pair NGC 5278/5279 (Arp 239) in its close vicinity, but that very galaxy was considered earlier as non-interacting with this merging pair [31]. UGC 1020 constitutes an isolated pair with a very small irregular dwarf PGC 212760—the mass difference between two is more than an order. Finally, UGC 7020A is involved in a wide pair with NGC 4081, both located at distant periphery of the group assembled around NGC 4125, in some 400 kpc from the group center. Hence, all the objects belong to loose groups, and none of them belong to clusters or rich groups. In the meantime, despite the very different visible environmental effects in their morphology, all four galaxies demonstrate intense gas exchange with their neighbors, as we have found in our study.

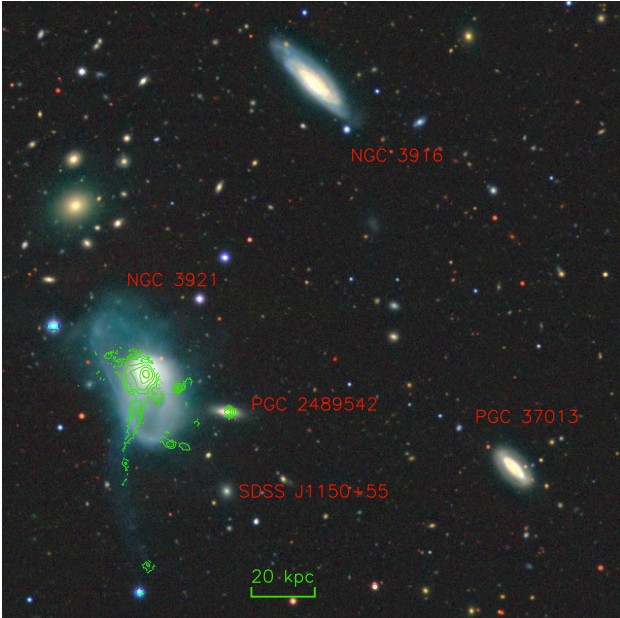 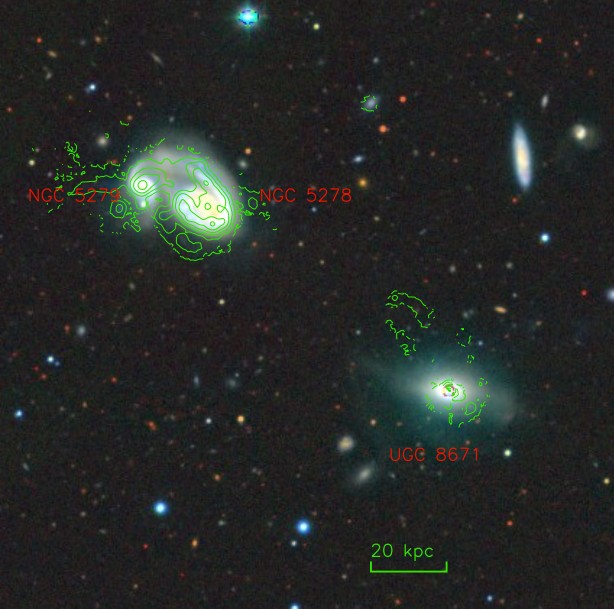

**Figure 1.** *Cont.*

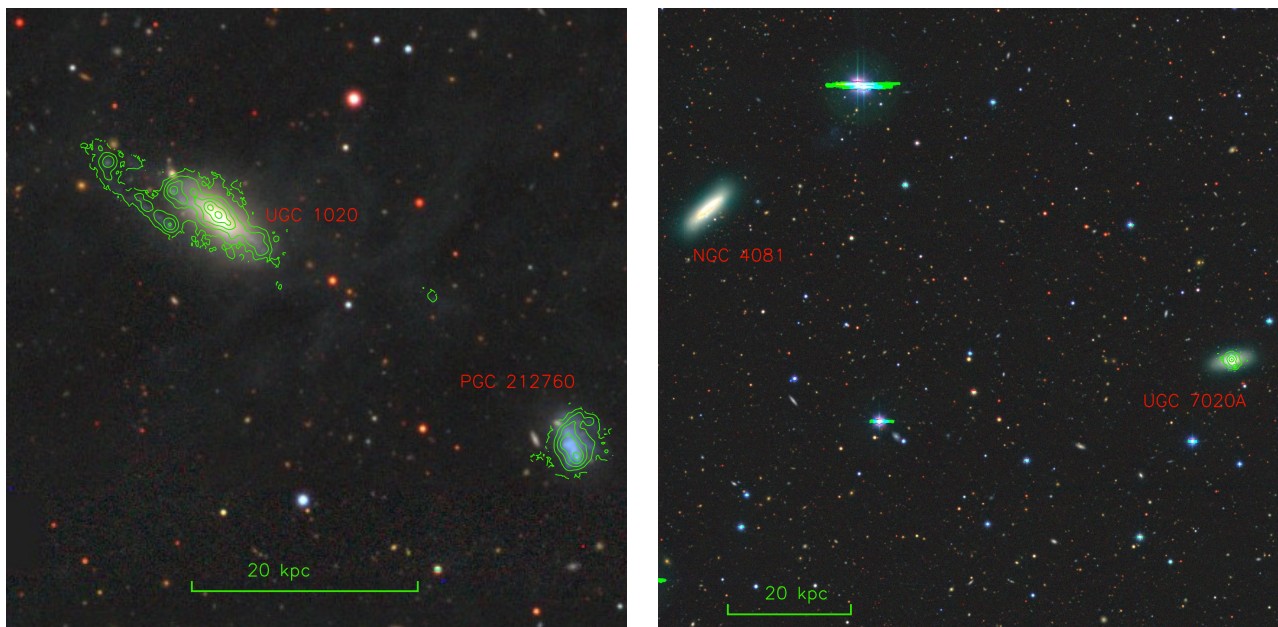

**Figure 1.** The images in combined colors (*grz*) for the galaxies studied in this work which are taken by us from the resource Legacy Survey [32]; the isophotes of the Hα images derived from the FPI data (see Figure 2) are overplotted.

**Table 1.** The galaxies studied and their neighbors.

| Galaxy and Its Distance | Type (NED [a]) | $M_K$ (LEDA [b]) | $M_B$ (LEDA) | $V_r$, km s$^{-1}$ (NED) | lg $M_*$, M$_\odot$ | Δ, kpc (NED) |
|---|---|---|---|---|---|---|
| **UGC 8671, D = 101 Mpc** | | | | | | |
| UGC 8671 | S0 (LEDA) | −24.6 | −21.35 | 7443 | | 0 |
| NGC 5278 | SA(s)b pec | −24.8 | −21.96 | 7569 | | 80 |
| NGC 5279 | SB(s)a pec | −23.7 | −22.06 | 7580 | | 98 |
| **NGC 3921, D = 81 Mpc** | | | | | | |
| NGC 3921 | (R′)SA(s)0/a pec | −24.7 | −21.82 | 5896 | | 0 |
| PGC 2489542 | E? S0? (LEDA) | −21.56 | −18.00 | 5707 | 9.82 [f] | 28 |
| SDSS J115059.3 + 550310 | S? (LEDA) | | −16.52 | 5865 | 9.03 [f] | 45 |
| NGC 3916 | SAb | −24.1 | −21.0 | 5786 | 10.84 [f] | 106 |
| PGC 37013 | Sa | −23.1 | −19.4 | 5494 | 10.58 [f] | 120 |
| **UGC 1020, D= 33.8 Mpc [c]** | | | | | | |
| UGC 1020 | Sab | −21.3 | −19.20 | 2589 | 9.63 [e] | 0 |
| PGC 212760 | Irr | | −16.17 | 2437 | 8.37 [e] | 34 |
| **UGC 7020A, D = 23.2 Mpc [d]** | | | | | | |
| UGC 7020A | S0 | −20.7 | −17.63 | 1515 | 9.08 [d] | 0 |
| NGC 4081 | Sa | −21.7 | −18.42 | 1408 | 9.75 [d] | 85 |

[a] NASA/IPAC Extragalactic Database, http://ned.ipac.caltech.edu (accessed on 12 October 2023). [b] Lyon-Meudon Extragalactic Database, http://leda.univ-lyon1.fr (accessed on 12 October 2023). [c] The Extragalactic Distance Database, http://edd.ifa.hawaii.edu (accessed on 12 October 2023). [d] [33]. [e] [34]. [f] [35].

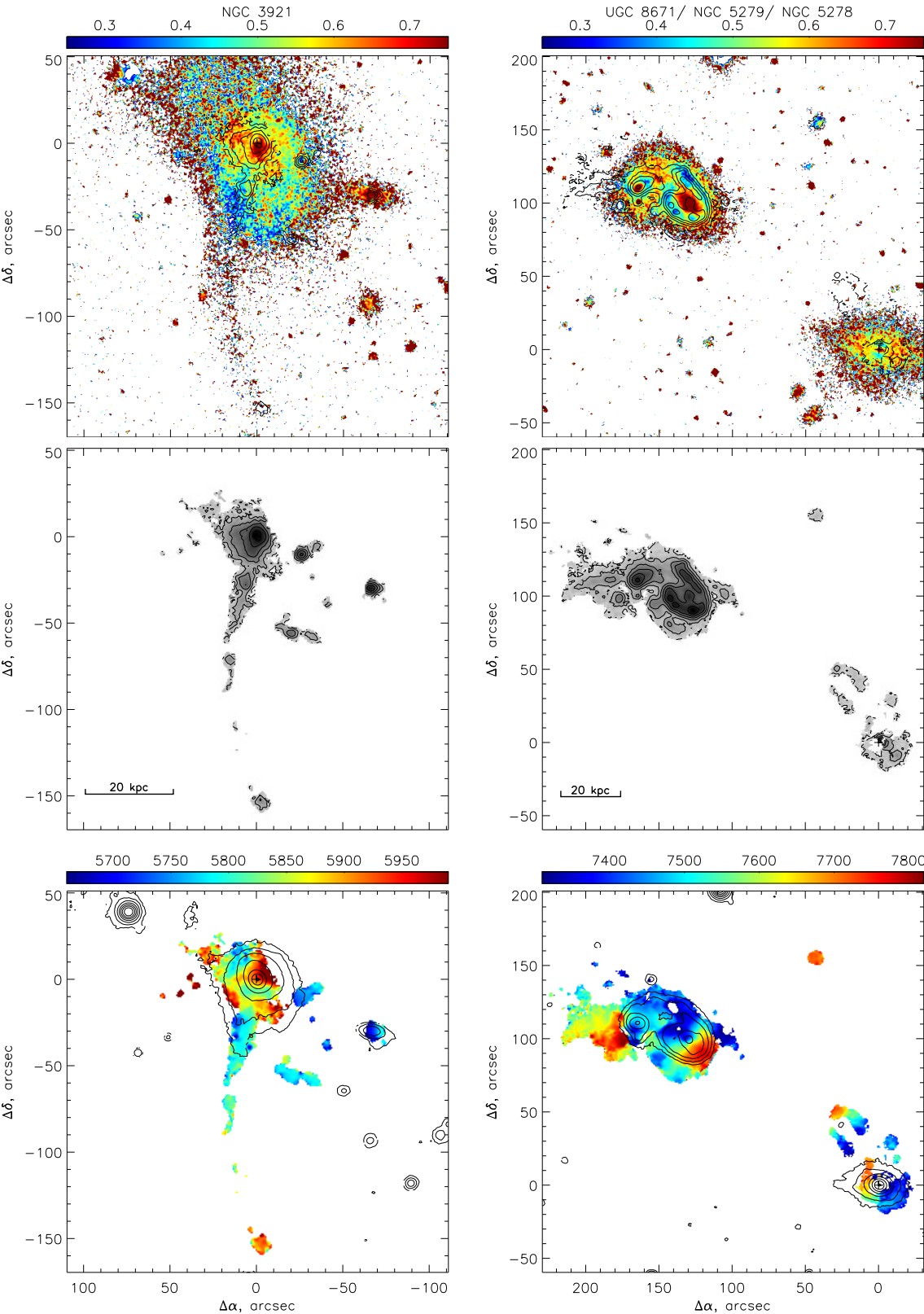

**Figure 2.** *Cont.*

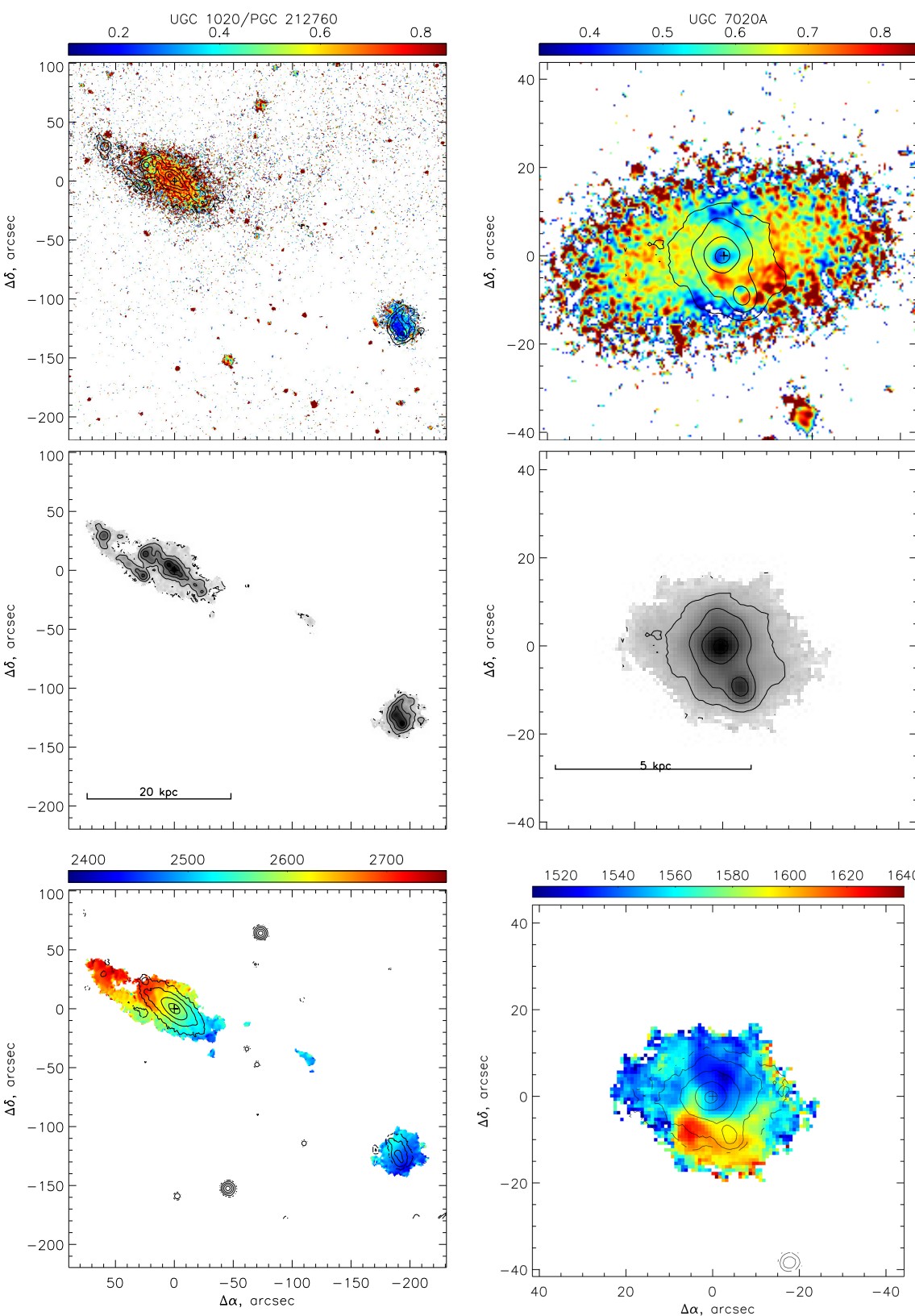

**Figure 2.** From top to bottom: the SDSS color index $(g − r)$ (smoothed with a Gaussian $FWHM = 1''$), the Hα flux map and velocity field derived from the FPI observations. Contours on the first two maps are Hα isophotes, whereas isolines on the velocity fields show the narrow-band continuum distributions derived from the same FPI data. The left panels present data for NGC 3921, the right ones for UGC 8671 and its companions.

### 3. Observations

We have undertaken a panoramic spectroscopy of our interacting galaxy systems by using the scanning Fabry–Perot interferometer (FPI, [36]) operating as a part of the SCORPIO-2 multi-mode reducer [37] at the prime focus of the 6 m telescope BTA of the Special Astrophysical Observatory, Russian Academy of Sciences (SAO RAS). Scanning FPIs make it possible to make a 3D spectroscopic study of the ionized-gas kinematics over the full field of view, whereas their spectral range usually only contains a single emission line selected by a filter with the band-pass FWHM of about 15–30 Å. The targets were observed at the SAO RAS 6 m telescope with the SCORPIO-2 in the FPI mode by using the interferometer providing a spectral resolution of $FWHM = 1.7$ Å ($\sim$78 km s$^{-1}$) and a full spectral range (interfringe) of $\Delta\lambda = 35$ Å. During the scanning process, we consequently obtained a few tens of interferograms by fixing different gaps between the FPI plates, to uniformly cover the interfringe: 40 frames in the vicinity of the red-shifted H$\alpha$ emission line. The field of view was 6.4 arcmin. The log of the observations including exposure times and mean seeing quality during the exposures is presented in Table 2. The data were reduced using the software package described in Moiseev [36]. After the primary reduction, airglow line subtraction, photometric and seeing corrections and wavelength calibration, the observed data were squeezed into data cubes, where to each spaxel of the 0.78 arcsec size a 40-channel spectrum was attributed. The data cubes were rotated to the "standard" orientation (N at the top, E to the left). The exact astrometry grid was created using the Astrometry.net project web-interface (http://nova.astrometry.net/ (accessed on 11 October 2023)). The optical continuum maps and monochromatic images in the emission line H$\alpha$, the line-of-sight velocity fields and velocity dispersion maps were created through one-component Voigt profile fitting of the spectra, as described in [38].

**Table 2.** Log of the scanning FPI observations.

| Galaxy | Date | Emission Line | Exposure, s | Seeing, $''$ |
|---|---|---|---|---|
| UGC 1020 | 6/7 October 2021 | H$\alpha$ | $40 \times 150$ | 3.0 |
| UGC 7020A | 20/21 May 2021 | H$\alpha$ | $40 \times 180$ | 2.7 |
| NGC 3921 | 27/28 March 2023 | H$\alpha$ | $40 \times 180$ | 4.2 |
| UGC 8671 | 27/28 March 2023 | H$\alpha$ | $40 \times 180$ | 4.4 |

Also for NGC 3921, we obtained a long-slit spectrum with the same multi-mode focal reducer SCORPIO-2 [37] at the prime focus of the BTA 6 m telescope. NGC 3921 was observed on 11 May 2021, setting the $1''$-slit close to the isophote major axis, $PA(slit) = 0$ deg, with the total exposure time of 2400 s ($4 \times 600$ s). The seeing during these observations was $FWHM \approx 2.5$ arcsec. We used the VPHG1200 grism with a maximum effectiveness at $\lambda \approx 5400$ Å providing an intermediate spectral resolution of $FWHM \approx 5$ Å (corresponding to the instrumental $\sigma$ of 130 km s$^{-1}$), to obtain a spectrum over the wavelength range from 3600 Å to 7200 Å. This spectral range contains a set of strong absorption and emission lines, making it suitable to analyze both stellar and gaseous kinematics of the galaxy and its resolved stellar populations. The slit was $6'$ in length, making it possible to use the edge spectra to subtract the sky background. The E2V CCD261-84 [39], with a format of $2048 \times 4096$ px, using the $1 \times 2$ binning mode, provided a spatial scale of 0.39 $''$/px and a spectral sampling of 0.9 Å/px.

For the photometric analysis, we have public data from the surveys SDSS/DR12 [40] and DECaLS [32].

### 4. Results

The results of measuring galaxies with the scanning Fabry–Perot interferometer include the mapping of the H$\alpha$ emission-line intensity and line-of-sight velocity fields of the ionized gas emitting in the H$\alpha$ line. We present these results in Figure 3 below and analyze them by comparing the H$\alpha$ intensity distributions with the distributions of young stellar populations traced by the blue broad-band color $g - r$, and we derive the parameters of the

gaseous disk spatial orientations by applying the tilted-ring model to the gas LOS velocity fields under the assumption of gas planar circular rotation, by using the original software DETKA [41]. Besides the inner kinematics of the gaseous disks within the galaxies, the velocity fields over the nearest galaxy outskirts also make it possible to trace outer gas tails and flows connecting the main galaxies with their neighbors.

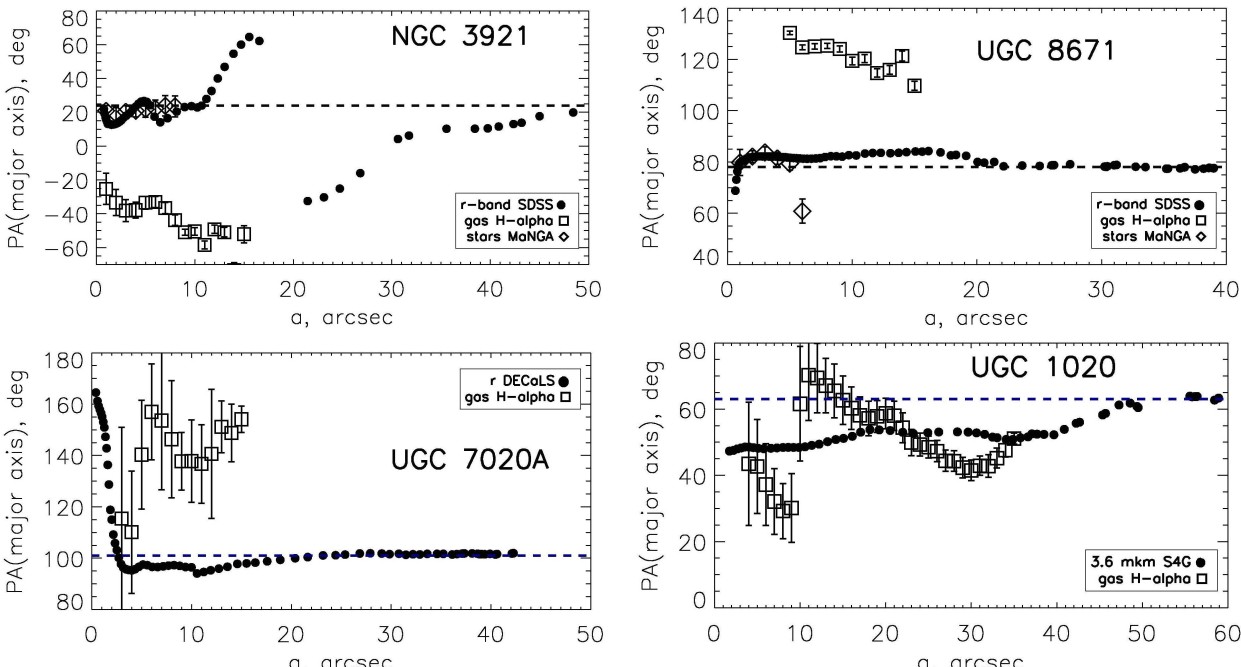

**Figure 3.** The comparison of the kinematical major axes position angles with the photometric lines of nodes in the galaxies studied.

Let us consider individual objects together with our interpretation of their Hα structures and velocity fields.

*4.1. NGC 3921*

The galaxy is a known merger [42]. It has been designated in the famous atlases of interacting galaxies as VV031 by Vorontsov-Velyaminov [16] and as Arp 224 by Arp [43]. However, contrary to major-merger model predictions, it has an exponential surface brightness profile from its stellar disk, though it is of the pressure-supported kind from a dynamical point of view [44].We also calculated the azimuthally averaged surface brightness profile for NGC 3921 by using the SDSS data. It is presented in Figure 4, together with its fit by an exponential law. Beyond $R > 50''$, it is a classical exponential disk, with the scale length of $16''$, or about 6 kpc. Closer to the center, in the radius range of $18''$–$35''$, the scale length is the same, namely $16''$ or 6.3 kpc, but the brightness is dimmer by some 0.5 mag per sq. arcsec, perhaps due to the dust concentration in the center of the galaxy. Therefore, the global structure of NGC 3921 gives evidence for it to be rather a product of minor merging which has not destroyed its large stellar disk. The north-eastern loop lacking's both ionized gas (Figure 2) and neutral hydrogen [45] may be stellar debris of a destroyed satellite.

We also obtained for NGC 3921 a spectrum with the SCORPIO-2 reducer in the long-slit mode, with the slit aligned along the north–south direction, and the gas and star line-of-sight (LOS) velocity profile is shown in Figure 5. The velocity profile for the stellar component is in full agreement with the earlier observations by Simien and Prugniel [46], though the latter spectral cut was made in a slightly different position angle, namely $PA = 20°$. The velocity profile for the ionized gas is different from that for the stellar component, implying that even the central gas in NGC 3921 is affected by the interaction

(merging). In Figure 3, upper left, the orientations of the kinematical major axes for the stellar component (which has been derived from the data of the MaNGA survey [47] in the last SDSS data release, SDSS/DR17 [48]) and for the ionized gas are strongly different. But the surprise is that the photometric major axis at radii $R > 20''$ continues the central kinematical major axis of the ionized gas smoothly. The central photometric major axis, at $R < 10''$, is in turn consistent with the orientation of the stellar rotation plane. We may conclude that NGC 3921, considered as a host galaxy, has conserved its central stellar component intact during the interaction, but the whole central gas content is accreted from the outside; at outer radii, the considerable fractions of stars are also acquired from outside, together with the absorbed satellite.

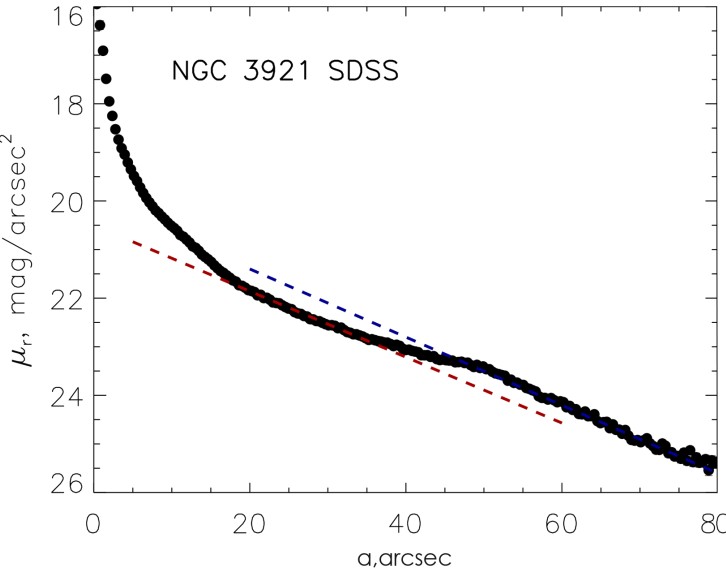

**Figure 4.** The surface brightness profile of NGC 3921, with the fitted exponential disks.

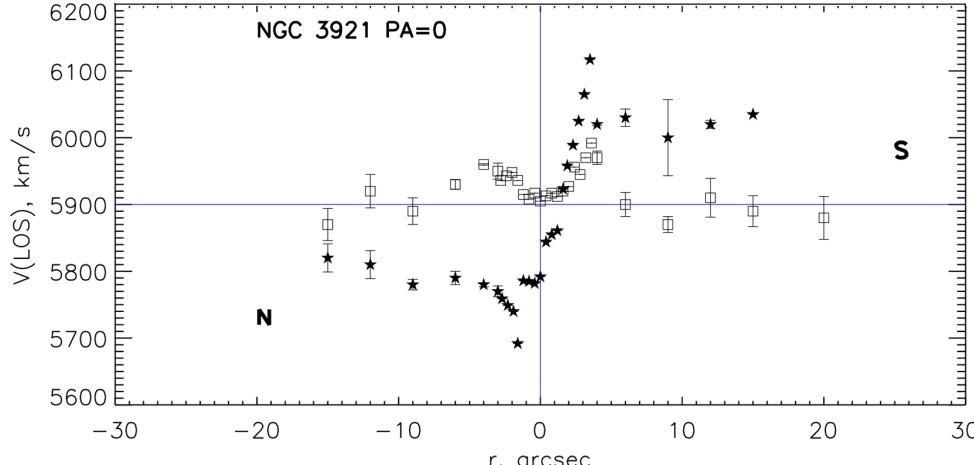

**Figure 5.** The line-of-sight velocity profiles for the stars (stars) and ionized gas (squares) in NGC 3921 along the north–south direction.

The long-slit spectrum in the full optical range allowed us to calculate fluxes of several strong emission lines along the radius of the galaxy and to probe the gas excitation mechanism. The so-called BPT-diagram [49], where the high-excitation emission line [OIII]$\lambda$5007 is confronted to some low-excitation lines, separates the excitation of the HII-region type (by young stars) from that by an active galactic nucleus or by shock fronts. In Figure 6, we analyze our measurements in NGC 3921 and its satellites by confronting

the measured emission-line flux ratios to the model of HII-regions from [50] and to the star-forming galactic nuclei sequence from [51], as well as to shock models by [52]. We found that the gas in the center of NGC 3921, within the radius of $R < 10''$ (or $R < 4$ kpc), is excited most probably by shock waves. In this part of the BPT-diagram, the shock excitation cannot be distinguished from that produced by a LINER-type active nucleus. We may prefer the shock excitation only because this area is far from the center, in 3–4 kpc off. The nucleus of NGC 3921 demonstrates the similar excitation at the BPT-diagram being the LINER-type active nucleus, as seen on the right of Figure 6. The only emission-line region caught by our long-slit spectrum which has the HII-type excitation is that in $153''$, to the south from the galaxy, which may be in fact a dwarf satellite—or a tidal dwarf [53]—at the end of the emission-line gaseous extended tail (Figure 2). Also in Figure 6, left, we include the data for the nucleus of PGC 2489542 which are taken from the SDSS survey. The nucleus of this dwarf galaxy is also "star-forming", or excited by young stars.

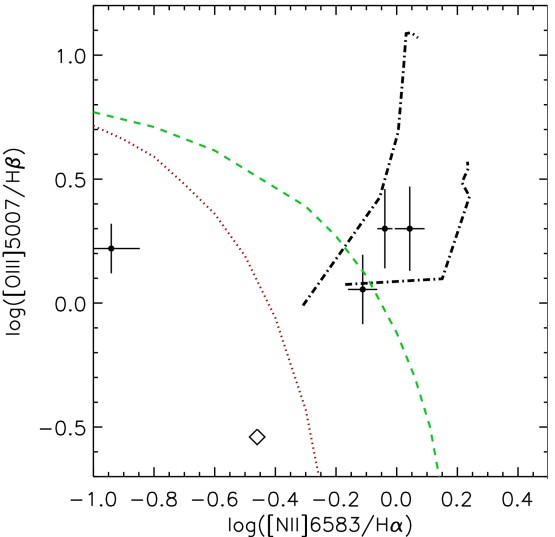 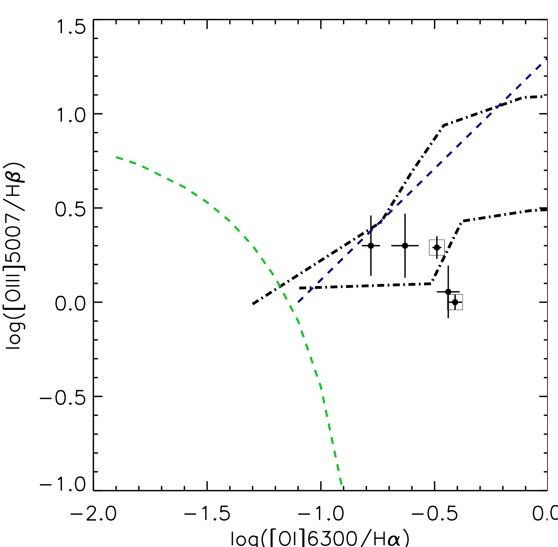

**Figure 6.** Baldwin–Phillips–Terlevich (BPT-) diagrams for the ionized gas in NGC 3921 along the north–south slit direction and in its satellites; the excitation mechanism's dividing lines are from [50] (the green dashed line) and from [51] (the red dotted line), and the dashed-dot lines show the shock excitation models by [52]. Two various diagrams are presented, with the circumnuclear measurements for NGC 3921 marked by the square frames shown only at the right plot: **left**—lg([OIII]$\lambda$5007/H$\beta$) vs. lg([NII]$\lambda$6583/H$\alpha$). The dots refer to NGC 3921's long-slit spectrum and the diamond shows the center of PGC 2489542 according to the SDSS data; and only the clump in $153''$ to the south from the center settles in the HII-region-like area; **right**—lg([OIII]$\lambda$5007/H$\beta$) vs. lg([OI]$\lambda$6300/H$\alpha$), wherein the blue dashed line [54] marks the boundary between Seyfert active galactic nuclei at the top and LINERs at the bottom.

For the emission-line gaseous clumps excited by young stars, we can calculate gas oxygen abundance by using popular strong emission-line calibrations. We used the calibrations of the indicators N2 and O3N2 from the paper by Pettini and Pagel [55]. Both indicators have given consistent estimates: in the southern clump, $12 + \log O/H = 8.36$, and in the nucleus of PGC 2489542, $12 + \log O/H = 8.70 \pm 0.05$. The nearly solar metallicity of the gas in the dwarf galaxy PGC 2489542 exceeds one expected for its luminosity (stellar mass): the known mass–metallicity relation, taken from, e.g., [56], promises $12 + \log O/H = 8.50$ for $\lg M_* = 9.82$ (Table 1). If we try to roughly estimate the mass of the southern tidal dwarf by inspecting the magnitude difference between PGC 2489542 and the tidal dwarf according to SDSS/DR12, $i = 15.54$ versus $i = 19.13$, we obtain $\lg M_* \approx 8.4$. Then, the metallicity expected from the mass–metallicity relation derived from Pettini and Pagel's

calibration for a large sample of galaxies by [57] is $12 + \log \mathrm{O/H} = 8.2$, which is also lower than the measured value $12 + \log \mathrm{O/H} = 8.36$.

The main result of our panoramic spectroscopy for the NGC 3921's field is the detection of ionized-gas flows beyond the stellar body of the host galaxy. In Figure 2, besides the prominent southern tidal tail containing the dwarf galaxy at its end, we see two linear H$\alpha$ flows to the west from the galaxy. The more northern one of them demonstrates the lower line-of-sight velocities than the bulk of the ionized gas in NGC 3921; this is consistent with being thrown away by PGC 2489542, or torn away by NGC 3921 from this satellite. The more southern one of them is directed toward the satellite SDSS J115059.3+550310, which represents a dwarf galaxy without current star formations; its spectrum obtained in the frame of the SDSS survey contains no emission lines but demonstrates a significant amount of prominent Balmer absorption lines that may make it possible to classify it as "E+A"—a galaxy with a recently quenched star formation. Since the line-of-sight velocities of the south-western ionized-gas filament are close to those of the main gas within NGC 3921 and since the systemic velocity of SDSS J115059.3 + 550310 is similar to the systemic velocity of NGC 3921, we may suggest that this second gaseous flow is the former gas content of SDSS J115059.3 + 550310, which is completely stripped away by the host galaxy. Both western gaseous filaments will probably be accreted by NGC 3921, being the main "gravitational attractor" in the field.

*4.2. UGC 8671*

This galaxy itself is not a member of any list of interacting galaxies; however, it looks like a regular early-type galaxy with low surface-brightness shells recognizable in deep images of the DECaLS survey. But in some 80 kpc to the north from it, there is a famous pair of strongly interacting galaxies Arp 239, consisting of NGC 5278 and NGC 5279. In the study by Repetto et al. [31], having undertaken a panoramic spectroscopy of Arp 239 with the scanning Fabry–Perot interferometer PUMA, the galaxy pair was classified as interacting galaxies of M51-type, not as a merger. The argument in favor of this conclusion was a regular character of the gas velocity fields in both NGC 5278 and NGC 5279, though with prominent radial motions. The earlier spectral study of Arp 239 with a long slit was undertaken by [58], and they had also classified this galaxy pair as interacting galaxies of M51-type.

The stellar mass of UGC 8671 is similar to the stellar mass of NGC 5278 as we can see from their K-magnitudes in Table 1—both are giant galaxies. So one cannot be very surprised to find signatures of gas transport between Arp 239 and UGC 8671. In Figure 2, we can see a short blue-shifted gaseous tail starting from the western part of NGC 5278 toward UGC 8671—if we suppose that UGC 8671 is closer to us than Arp 239, which may be implied by the smaller systemic velocity of UGC 8671 with respect to NGC 5278. As for the (past) gas content of UGC 8671, it has abandoned the central part of the galaxy and represents now a large loop with the northern red-shifted tip—in agreement with the background position of NGC 5278. Its kinematical major axis, seen in the upper right of Figure 3, has nothing in common with the isophote (stellar disk) orientation of UGC 8671—perhaps the model of circular gas rotation is inapplicable in this case.

If we look at the present structure of UGC 8671, we see a lenticular galaxy with a large bulge, though its morphological type is somewhat uncertain: according to NED, it is S?, and Repetto et al. [31] posit that it is an edge-on elliptical. Figure 7 shows the surface brightness profile of UGC 8671 calculated by us from the SDSS data; the exponential disk starts to dominate only at $R > 20''$, or farther than 10 kpc from the center. The galaxy demonstrates a very high concentration of the light at the center: its effective radius is $3''$, according to the 2MASS survey data [59], or 1.5 kpc. Meanwhile, the size-luminosity scaling relation (e.g., [60]) promises an effective radius of more than 10 kpc for the absolute magnitude which is measured for UGC 8671 (Table 1). This inconsistency implies that the stellar body of UGC 8671 may be strongly stripped by the interaction.

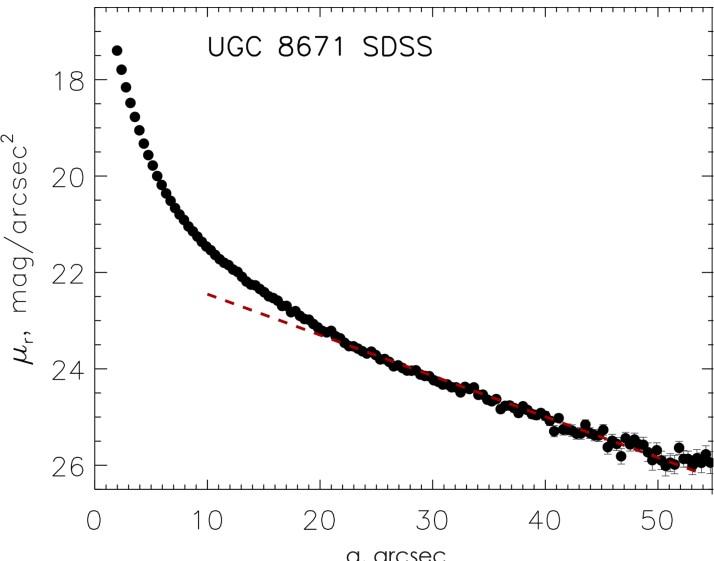

**Figure 7.** The surface brightness profile of UGC 8671, with the fitted exponential disk.

### 4.3. UGC 1020

This galaxy is a member of the tight pair with the dwarf PGC 212760; the galaxy stellar mass ratio in the pair is 20:1. In the Hα emission flux distribution, a gaseous bridge between two galaxies is clearly seen (Figure 2). If we consider the northern loop in the host galaxy as a continuation of this bridge and if we inspect the line-of-sight gas velocities in this loop+bridge system (Figure 2), we come to the conclusion that the velocity variations along the loop+bridge system reflect the rotation of the main galaxy.

Figure 8 shows the residual gas velocities inside UGC 1020 after subtracting the circular rotation model from the observed LOS velocity field. The largest residual (non-circular) velocities are demonstrated by the northern loop: from the geometrical point of view, this gas flow may just come to the galaxy at the western half and rise over the stellar disk with its eastern half. But one can also see residual velocities as high as 30 km s$^{-1}$ localized at the major axis of UGC 1020. This means that these residuals cannot be *radial motions in the disk plane*: the projection of radial velocities within the disk plane onto our line of sight is maximal at the minor axis and zero at the major axis (at the line of nodes). The localization of residual velocities near the major axis means that we deal with off-plane gas motions. The behavior of the $PA_{kin}$ radial profile in Figure 3 represents variations around the photometric major axis (line of nodes) between $R \approx 5''$ and $R \approx 35''$; this may be consistent with probable ionized-gas flow vertical precession with respect to the stellar disk plane.

The literature data on the neutral gas content in UGC 1020 and PGC 212760 imply almost equal HI amounts in both lg $M(HI) = 9.49$ and lg $M(HI) = 9.41$, respectively [61], despite the difference in the stellar masses by a factor of 20 (the ALFALFA survey has a spatial resolution sufficient to separate UGC 1020 and PGC 212760). Evidently, the rather large UGC 1020 has already shared its gas content with its small satellite. There are already similar cases of large galaxies providing a gas supply for their small early-type neighbors in the literature. For example, the study of Arp 70 by Rampazzo et al. [62] by means of Fabry–Perot interferometry allowed them to elaborate conclusions on the outer gas origin in the small PGC 212740 provided by gas inflows onto it from UGC 934 (so-called "cross-fueling effect").

However, taking into account that PGC 212760 is off the scaling relation connecting stellar and gaseous masses of galaxies according to, e.g., [63], the fact that it is much higher and the fact that UGC 1020 surpasses also the median trends in [64], it seems more probable to us that two galaxies have perhaps a common extended HI disk, similar, say, to the

common extended HI disk of the isolated pair IC 719 and IC 718 [65–67]. The ionized-gas flow between two galaxies visible in Figure 2 is then perhaps a consequence of the large HI disk tidal distortion provoked by a close orbital passage of PGC 212760 near the host galaxy.

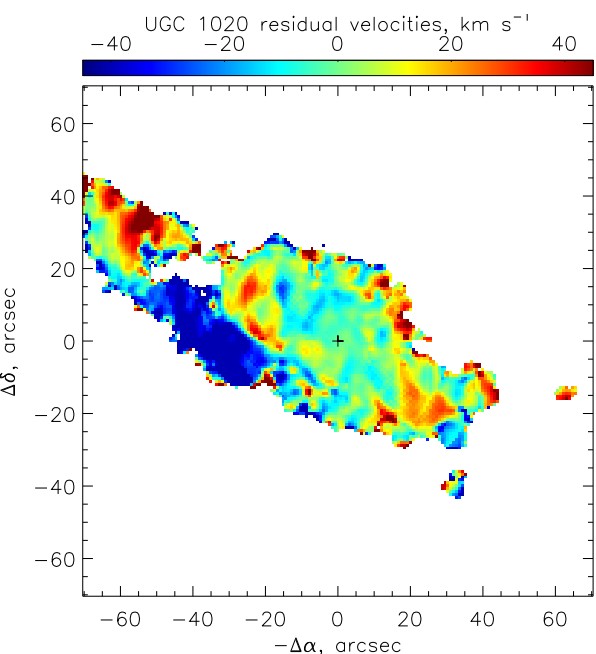

**Figure 8.** The residual gas velocities inside UGC 1020, after subtraction of the circular rotation model.

### *4.4. UGC 7020A*

This early-type galaxy has no outer morphological features, which may be seen as signatures of possible galaxy interaction, despite the existence of a rather close and more massive neighbor, NGC 4081 (Table 1). But it has a clearly visible inclined gaseous disk with a current star formation within it (Figure 2). By applying our tilted-ring analysis with the software DETKA [41], we determined not only the position angle of the line of nodes for the gaseous disk, but also its inclination equal to $62°$ in the radius range of $7''–15''$. By assuming that the stellar disk is inclined by some $70°$ that is implied by the maximum isophote ellipticity of 0.58, we obtain, within the morphological uncertainties, the angle between the stellar and gaseous planes equal to $46°–68°$ or $92°–114°$, whether the northern or southern part of the disk is the nearest to us. So it is just a *highly inclined* gaseous disk, or may even be a polar one.

Recently, Chen et al. [68] have presented their study of 2D kinematics in UGC 7020A applying optical integral-field spectroscopy and millimeter-wave radio-interferometry. Their stellar velocity field for UGC 7020A demonstrates a regular rotation in agreement with the photometric major axis. The ionized gas velocity field obtained by [68] has a similar pattern with our FPI data, while the FPI velocity field provides a larger spatial coverage and angular resolution. The kinematics of molecular gas (CO) is similar to our results for the H$\alpha$ emission line: the gas $PA_{kin}$ in the central few arcsec is close to the photometric $PA$, whereas a sharp turn of isovelocities occurs at $R \geq 5''$. We can suggest that the inclined/polar gaseous structure is a ring or a disk with a central hole, while in the very center we see circumnuclear gas rotating in the stellar disk plane.

Unlike the conclusion by Chen et al. [68], we cannot claim with certainty that the inner compact clump in some $10''$ to the south-west from the galaxy nucleus is a remnant of a minor merger related to this inclined gaseous disk. The clump does not project against the blue, strongly inclined disk (Figure 2), and its line-of-sight velocity is higher than predicted by the circular rotation model of the gaseous disk of UGC 7020A for the clump location. The color of the clump is rather red (Figure 2), though the prominent UV radiation is detected in this location. Perhaps, it could be argued that the event having provided the gas supply

for UGC 7020A and the minor merger whose remnant is seen against the south-western clump are rather different events.

## 5. Discussion

We examined the emission line Hα, radiated by ionized hydrogen, in the close environments of four interacting galaxy systems by applying the panoramic spectroscopy approach. All four systems are very compact, with the neighboring galaxies inhabiting within 100 kpc from the target objects.

We have found a few cases where we may conclude that the larger galaxy attracts the material of a smaller galaxy (NGC 3921 and its closest western satellite) and even merges it (UGC 7020A). This is just what has been expected as one of the main scenarios of a disk galaxy feeding from the outside. However, the whole picture is not so simple.

In three of four systems, except the smallest low-massive UGC 7020A, we have found ionized-gas filaments (flows) beyond the main bodies of the galaxies under consideration. Sometimes, the LOS velocities of these flows show their relation with satellites; but the direction of the flows implies their possible future accretion onto the host galaxies. It seems probable that we observe accreting gas inflows which are widely expected to support the evolution of disk galaxies. However, in some cases, especially in the galaxy pair UGC 1020+PGC 212760, we do not find decisive arguments in favor of gas transport from one galaxy to another; even taking into consideration substantial mass difference, as in the system of UGC 1020 and its small satellite PGC 212760, the gas velocity distribution does not make it possible to determine with certainty that the intergalactic gas flows come from the small galaxy and into the larger one. Conversely, there are hints that the compact galaxy groups may possess common gas reservoirs which happen to be disturbed and to produce gas inflows onto both galaxies when the galaxies within it move closer to each other. But the close mutual passages must also disturb the inner gaseous disks of interacting galaxies, and they may lose their gas to deliver it back into the intergalactic gas reservoirs. The latter situation is probably now observed in NGC 3921 and in the Arp 239+UGC 8671 system. In NGC 3921, a large HI ring is observed [45], with the ring center shifted by some 1 arcmin to the south from the galaxy center. The velocity field of the HI ring implies a rotation spin consistent with the central rotation of the stellar component, so it may be a full former gas content of the NGC 3921 disk stripped away by the interaction.

We may consider the possibility that the feeding from the common gas reservoir may also be one of the possible sources providing the disk galaxy's smooth evolution. The argument in favor of this hypothesis would be the correct positioning of the galaxies with the common gas reservoirs at the scaling relations connecting the gas content and star formation rate with the stellar masses of the galaxies. In Table 3, we have assembled a small sample of galaxy pairs embedded into a common gas cloud: NGC 5713 + NGC 5719 [23], IC 719 + IC 718 [65] and NGC 128 + NGC 127 [69]; in the latter case, the HI cloud is small, and it does not embrace the whole galaxy system but is located between two galaxies. Figure 9 presents the scaling relations taken from the recent data of the xGASS survey [64,70] for large galaxies and from the WHISP + WISE survey for dwarf galaxies [63]. In the lg $M(HI)$ vs. lg $M_*$ plot, we do not see any asymmetry between the members of the galaxy pairs: both IC 719 and IC 718 belong to the main (median) trend, and both NGC 5713 and NGC 5719 (and both UGC 1020 and PGC 212760) lie in the gas-rich area of the plot. Meanwhile, if one member of a galaxy pair is a donor and other member is a recipient of the gas, the former must look like a gas-deficient galaxy. We do not observe such an effect. Two galaxies, namely NGC 127 and NGC 128, both being gas-deficient (the latter has no neutral hydrogen detected at all) confirms the whole tendency. The absence of asymmetry between two pair members at the lg $M(HI)$ vs. lg $M_*$ plot gives evidence in favor of equal probabilities to obtain gas from the common cloud for every galaxy, independently of its mass.

**Table 3.** The galaxy pairs with common gas envelopes.

| Galaxy | Type (NED [a]) | lg $M(HI)$, $M_\odot$ | lg $M_*$ [b], $M_\odot$ | lg $SFR$ [b], $M_\odot$ per Year |
|---|---|---|---|---|
| NGC 5713 | SAB(rs)bc pec | 9.99 [c] | 10.00 | +0.49 |
| NGC 5719 | SAB(s)ab pec | 9.92 [c] | 10.32 | −0.06 |
| IC 719 | S0? | 9.01 [d] | 10.10 | −0.67 |
| IC 718 | Im? | 9.08 [d] | 9.01 | −1.18 |
| NGC 127 | SA0$^0$ | 7.98 [e] | 9.94 | −0.09 |
| NGC 128 | S0 pec | | 11.22 | −0.08 |

[a] NASA/IPAC Extragalactic Database, http://ned.ipac.caltech.edu (accessed on 12 October 2023). [b] [33]. [c] The Extragalactic Distance Database, http://edd.ifa.hawaii.edu (accessed on 12 October 2023). [d] [64]. [e] [69].

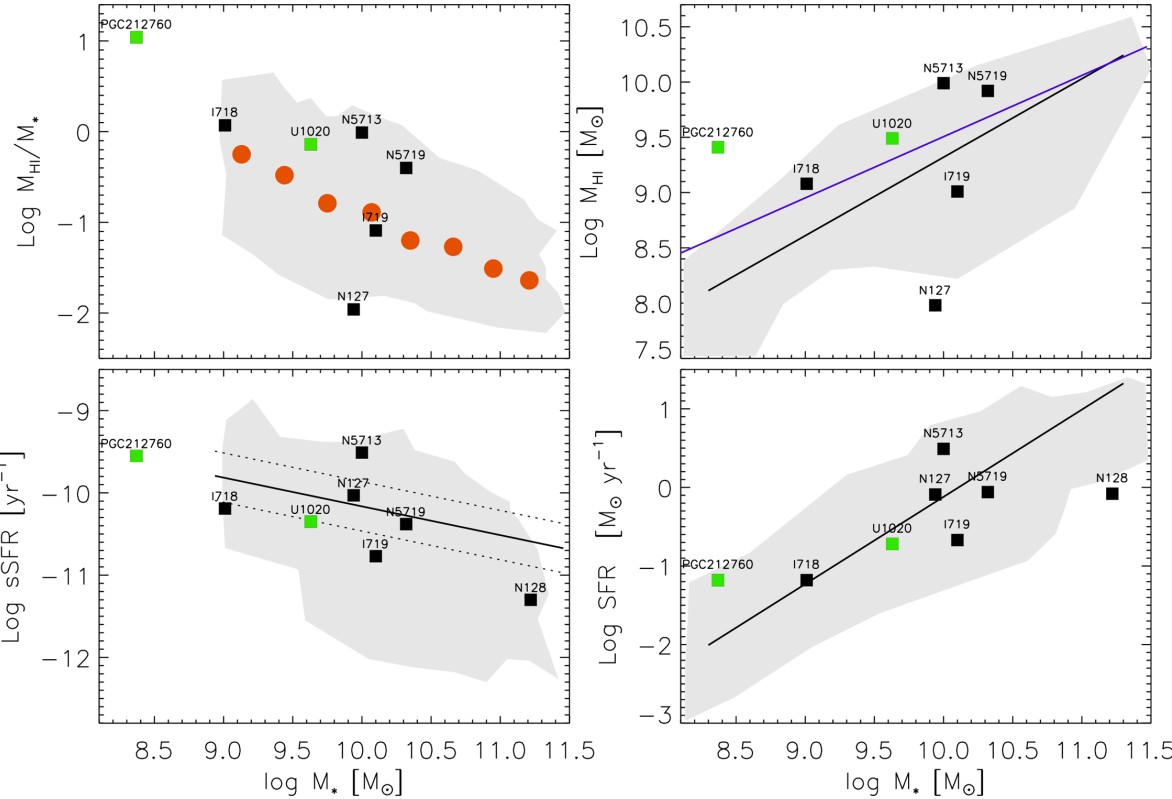

**Figure 9.** The scaling relation positions of the group galaxies accreting cold gas from the common hydrogen clouds. The gray shadow regions mark the scatter of data in the scaling relations from the original papers, and the best-fit lines are also present in the works cited below. **Left panels**: the HI vs. $M_*$ relation [64] and the main sequence $sSFR$ vs. $M_*$ [70], both basing on the xGASS survey. The red points indicate weighted average values, the solid line is a main sequence with ±0.3 dex borders (the dotted lines). In the **right column**, the same scaling relations for dwarf galaxies are from [63] (the WHISP survey). The black line indicates the best fit to the $M(HI)$ contained within the stellar disks, whereas the magenta line corresponds to the total $M(HI)$. The squares show the positions of the galaxies from Table 3 (black) and, for comparison, the characteristics of the UGC 1020 and PGC 212760 (green squares).

As for star formation, here we see a rather inhomogeneous picture: half of the galaxies—members of these pairs, having the morphological types from S0 to Irr—are in the green valley, while NGC 5719, PGC 212760 and NGC 127 (a gas-poor one) are just at the main sequence, and NGC 5713 demonstrates a starburst. We think that the different star formation regimes under a similar rate of gas inflow may be related to the geometry of gas accretion. In our small sample, we see the clearly inclined direction of gas inflow

in UGC 1020 (this work), and the in-flowing gas kinematics in agreement with the stellar kinematics in NGC 127 [71].

The possibility of different geometry of the gas inflow onto disk galaxies is, in our opinion, a very promising factor shaping the disk galaxy morphology. Indeed, the origin of S0 galaxies in the field, including loose groups, is still a puzzle because the commonly accepted environmental effects, such as hot intergalactic gas ram-pressure, cannot quench star formation in their disks when a galaxy inhabits a rarefied environment. Meanwhile, S0 galaxies in the field (which are a majority of the S0 population) often demonstrate decoupling of their stellar and gaseous kinematics as a rather common feature [72–74]. In our present sample, we clearly see such decoupling in NGC 3921 and UGC 7020A. Their gas comes from directions which lie well off the stellar disk planes. And the gas inflow from an inclined direction may indeed suppress star formation within this acquired gas [75] due to the produced shocks and heating. So the origin of disk galaxy morphology may be related to the geometry of the outer gas arrival.

## 6. Conclusions

By using panoramic spectroscopy by means of the Fabry–Perot interferometry at the Russian 6m telescope, we have undertaken the study of ionized-gas kinematics and distribution inside and around several systems of interacting galaxies. Our sample included four tightly interacting systems of galaxies having different luminosities and stellar masses; all systems belong to loose groups. Our aim was to search for any signatures of gas transfer from satellites to the host galaxies, in the frame of the popular paradigm on the outer cold gas accretion governing disk galaxy evolution. We have indeed detected intergalactic ionized-gas filaments in three of the four galaxy systems studied. However, we cannot definitively state that, in these three cases, the small galaxies are the sources, and the large galaxies are the recipients of this intergalactic gas. The hypothesis of "cross-fueling" looks more attractive: either the mass ratio does not determine the direction of gas inflow, and the host galaxy may share its gas with a satellite, or two (or more) galaxies are embedded into a common huge gaseous cloud which serves as a reservoir of outer gas accretion for all of them.

**Author Contributions:** Conceptualization, O.S. and A.V.M.; Observations, R.U.; Data reduction, A.S.; methodology, A.V.M.; software, A.V.M.; formal analysis, O.S.; writing—original draft preparation, O.S.; writing—review and editing, A.V.M.; funding acquisition, O.S. All authors have read and agreed to the published version of the manuscript.

**Funding:** This research was funded by the grant No. 075–15–2022–262 (13.MNPMU.21.0003) of the Ministry of Science and Higher Education of Russian Federation.

**Institutional Review Board Statement:** Not applicable.

**Informed Consent Statement:** Not applicable.

**Data Availability Statement:** The data presented in this study are available upon request from the corresponding author. The data are not publicly available due to restrictions of the 6m telescope data archive.

**Acknowledgments:** We thank E. Malygin and E. Shablovinskaya for performing the long-slit spectroscopy of NGC 3921. This work is based on the data obtained at the Russian 6 m telescope of the Special Astrophysical Observatory carried out with the financial support of the Ministry of Science and Higher Education of the Russian Federation and on the public data of the SDSS and DESI surveys. Some broad-band photometric images were taken from the Legacy Survey collection. The Legacy Surveys consist of three individual and complementary projects: the Dark Energy Camera Legacy Survey (DECaLS; Proposal ID #2014B-0404; PIs: David Schlegel and Arjun Dey), the Beijing–Arizona Sky Survey (BASS; NOAO Prop. ID #2015A-0801; PIs: Zhou Xu and Xiaohui Fan) and the Mayall z-band Legacy Survey (MzLS; Prop. ID #2016A-0453; PI: Arjun Dey). DECaLS, BASS and MzLS together include data obtained, respectively, at the Blanco telescope, Cerro Tololo Inter-American Observatory, NSF's NOIRLab; the Bok telescope, Steward Observatory, University of Arizona; and the Mayall telescope, Kitt Peak National Observatory, NOIRLab. Pipeline processing and analyses of the data

were supported by NOIRLab and the Lawrence Berkeley National Laboratory (LBNL). The Legacy Surveys project is honored to be permitted to conduct astronomical research on Iolkam Du'ag (Kitt Peak), a mountain with particular significance to the Tohono O'odham Nation. NOIRLab is operated by the Association of Universities for Research in Astronomy (AURA) under a cooperative agreement with the National Science Foundation. LBNL is managed by the Regents of the University of California under contract to the U.S. Department of Energy. This project used data obtained with the Dark Energy Camera (DECam), which was constructed by the Dark Energy Survey (DES) collaboration. Funding for the DES Projects has been provided by the U.S. Department of Energy, the U.S. National Science Foundation, the Ministry of Science and Education of Spain, the Science and Technology Facilities Council of the United Kingdom, the Higher Education Funding Council for England, the National Center for Supercomputing Applications at the University of Illinois at Urbana-Champaign, the Kavli Institute of Cosmological Physics at the University of Chicago, Center for Cosmology and Astro-Particle Physics at the Ohio State University, the Mitchell Institute for Fundamental Physics and Astronomy at Texas A&M University, Financiadora de Estudos e Projetos, Fundacao Carlos Chagas Filho de Amparo, Financiadora de Estudos e Projetos, Fundacao Carlos Chagas Filho de Amparo a Pesquisa do Estado do Rio de Janeiro, Conselho Nacional de Desenvolvimento Cientifico e Tecnologico and the Ministerio da Ciencia, Tecnologia e Inovacao, the Deutsche Forschungsgemeinschaft and the Collaborating Institutions in the Dark Energy Survey. The Collaborating Institutions are Argonne National Laboratory, the University of California at Santa Cruz, the University of Cambridge, Centro de Investigaciones Energeticas, Medioambientales y Tecnologicas-Madrid, the University of Chicago, University College London, the DES-Brazil Consortium, the University of Edinburgh, the Eidgenossische Technische Hochschule (ETH) Zurich, Fermi National Accelerator Laboratory, the University of Illinois at Urbana-Champaign, the Institut de Ciencies de l'Espai (IEEC/CSIC), the Institut de Fisica d'Altes Energies, Lawrence Berkeley National Laboratory, the Ludwig Maximilians Universitat Munchen and the associated Excellence Cluster Universe, the University of Michigan, NSF's NOIRLab, the University of Nottingham, the Ohio State University, the University of Pennsylvania, the University of Portsmouth, SLAC National Accelerator Laboratory, Stanford University, the University of Sussex and Texas A&M University. BASS is a key project of the Telescope Access Program (TAP), which has been funded by the National Astronomical Observatories of China, the Chinese Academy of Sciences (the Strategic Priority Research Program "The Emergence of Cosmological Structures" Grant #XDB09000000), and the Special Fund for Astronomy from the Ministry of Finance. The BASS is also supported by the External Cooperation Program of Chinese Academy of Sciences (Grant #114A11KYSB20160057), and Chinese National Natural Science Foundation (Grant #12120101003, #11433005). The Legacy Survey team makes use of data products from the Near-Earth Object Wide-field Infrared Survey Explorer (NEOWISE), which is a project of the Jet Propulsion Laboratory/California Institute of Technology. NEOWISE is funded by the National Aeronautics and Space Administration. The Legacy Surveys imaging of the DESI footprint is supported by the Director, Office of Science, Office of High Energy Physics of the U.S. Department of Energy under Contract No. DE-AC02-05CH1123, by the National Energy Research Scientific Computing Center, a DOE Office of Science User Facility under the same contract; and by the U.S. National Science Foundation, Division of Astronomical Sciences under Contract No. AST-0950945 to NOAO. Some galaxy images are taken from the SDSS survey public database. Funding for the SDSS-III has been provided by the Alfred P. Sloan Foundation, the Participating Institutions, the National Science Foundation and the U.S. Department of Energy Office of Science. The SDSS-III website is http://www.sdss3.org/ (accessed on 12 October 2023). SDSS-III is managed by the Astrophysical Research Consortium for the Participating Institutions of the SDSS-III Collaboration including the University of Arizona, the Brazilian Participation Group, Brookhaven National Laboratory, Carnegie Mellon University, University of Florida, the French Participation Group, the German Participation Group, Harvard University, the Instituto de Astrofisica de Canarias, the Michigan State/Notre Dame/JINA Participation Group, Johns Hopkins University, Lawrence Berkeley National Laboratory, Max Planck Institute for Astrophysics, Max Planck Institute for Extraterrestrial Physics, New Mexico State University, New York University, Ohio State University, Pennsylvania State University, University of Portsmouth, Princeton University, the Spanish Participation Group, University of Tokyo, University of Utah, Vanderbilt University, University of Virginia, University of Washington and Yale University. We acknowledge the usage of the HyperLEDA database (http://leda.univ-lyon1.fr (accessed on 12 October 2023)). This research has made use of the NASA/IPAC Extragalactic Database (NED) which is operated by the Jet Propulsion Laboratory, California Institute of Technology, under contract with the National Aeronautics and Space Administration.

**Conflicts of Interest:** The authors declare no conflict of interest. The founders had no role in the design of the study; in the collection, analyses, or interpretation of data; in the writing of the manuscript; or in the decision to publish the results.

## Abbreviations

The following abbreviations are used in this manuscript:

| | |
|---|---|
| SAO RAS | Special Astrophysical Observatory of the Russian Academy of Sciences |
| FPI | Fabry–Perot Interferometer |
| SFR | Star Formation Rate |
| LOS | Line-of-Sight (Velocities) |
| BPT | Baldwin–Phillips–Terlevich (diagram) |
| SDSS | Sloan Digital Sky Survey |
| 2MASS | Two-Micron All Sky Survey |
| xGASS | GALEX Arecibo SDSS Survey |
| WHISP | Westerbork survey of H I in Irregular and Spiral galaxies |

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
