# Peer review of "S0 Galaxies: Outer Gas Accretion through Tidal Interaction and Minor Merging"

_galaxies, doi:10.3390/galaxies11060119_

Round 1
Reviewer 1 Report
Comments and Suggestions for Authors
This paper clearly presents the recent results of four merger galaxies, and try to study the gas accretion. This topic is crucially important to understand the galaxy evolution. I have several comments below:
1, Explain more about how the sample is selected, and how special is this sample comparing with other galaxy pair studies, such as pairs selected from e.g., MANGA or other projects. And what are the advantages of this sample. This is important to understand how representative are the results.
2, Add one section of conclusion to the end of this work, especially how will this study help to understand the gas accretion. The current paper only list the observational results, with no conclusion.
3, Add scale bar in Fig. 1. Show NGC4081 in the forth panel.
4, Explain what is the thick black dots or circle or circles in Fig. 2, e.g., the black thick circle at (-30, 15) of NGC3921; two black dots at about (20,24).
5, Bottom panel of Fig. 2 is too small to see.
Comments on the Quality of English Language
The author should check the spelling, such as the FHWM in page 4 should be FWHM.
Author Response
The answers to the referee's comments are listed in the PDF file attached. The changes in the text are shown by boldface.

Reviewer 2 Report
Comments and Suggestions for Authors
The paper S0 GALAXIES: OUTER GAS ACCRETION THROUGH TIDAL INTERACTION AND MINOR MERGING, by O. Sil'chenko et al. presents new data from Fabry-Perot observations, yielding high spectral resolution information of four galaxies belonging to interacting systems. The Fabry-Perot observations are difficult and time consuming to obtain, so that they deserve to be published at any rate. However, the paper is based on only 4 systems that are rather different among themselves. Here I list several issues and suggestions that the authors should consider, in my opinion, to make the paper suitable for acceptance.
* English needs extensive corrections. Some words are used with a meaning different from the intended one.
* Fig. 1: identify clearly the sources, add scale bar with arcsec and projected linear distance, and orientation. "SDSS..." or LEDA... in Fig. 10 are not a proper identification.
* I suggest that the authors make an effort to straighten the presentation of the data. The g-r color maps are helpful but are used only in the analysis of a detail in one of the sources. The narrow band continuum in Fig. 3 is barely visible and provides no clear information.
* Overlay the isophotal contours with the intensity map of Halpha over the image. This will give to the reader an immediate perception of the differences/similarities between the stellar and gaseous component emission, combining Fig. 1 and the right images of Fig. 2. Detection of extended filaments joining the galaxies in the pair or at least beyond the stellar body is really the main result and should be adequately made clear in the Figure stressed in the discussion.
* Join side by side the velocity maps of Fig. 3 to the color images with the overlaid Halpha intensity maps in a new Figure 1. Please try to show the two side-by-side panels for each galaxy with the same angular size.
* NGC 3921: there are ring like structures that may suggest one or multiple crossings. Perhaps LEDA 2489542 is a culprit? This hypothesis should be mentioned.
* Fig. 7: I agree with the authors that shock is a likely perspective. But how can they be sure there is shock? A LINER-like nuclear spectrum could be associated with photoionization at low ionization level. The extended gas might be photoionized by an active nucleus. Please discuss this point.
* UGC 8671, p. 20: the first paragraph delves with a pair of galaxies 80 kpc away from UGC 8671; it is OK to mention it but only after the properties of UGC 8671 are explained.
* "In the nearest future Arp 239 and UGC 8671 would be probably connected by a gaseous bridge." In the nearest future? Specify a time scale. The trail of emission leads to think that some tidal stripping between Arp 239 and UGC 8671 might have already occurred.
* Some references on cross-fueling of galaxies might be added. A search in NASA-ADS revealts that the authors are not the first to consider mass transfer across galaxies.
* None of the systems apparently belong to rich cluster, but this is not said explicitly in the paper.
* Fig. 10 should be improved. The two galaxies of the present study should by plotted with a different symble.
* The discussion on LEDA 212760 and UGC 1020 in Section 5 appears appropriate, and fairly interesting. However, the sample is heterogenous and is selected only on the basis of the resemblance of one of the components with an S0 galaxy. The authors may gain some further general conclusions. For instance, NGC 3921 is clearly not an S0. Some considerations could be added on the evolution of the S0 galaxies, and on the prevalence of them in interacting systems and in cluster, to address the issue whether the systems of the present work are fairly normal, or are rare, or even extremely peculiar.
Comments on the Quality of English Language
* English needs extensive corrections. Some words are used with a meaning different from the intended one.
* Many typos. Examples: Either... nor, scenaria, fueled, squized, intacted, etc.
Author Response
The answers to the referee's comments are included into the PDF file attached. The changes in the manuscript are enhanced by boldface.

Round 2
Reviewer 1 Report
Comments and Suggestions for Authors
I would like to thank the authors for considering my previous comments. The results are now clearer to me. I have one more concern about the 'accretion flows' mentioned in the abstract:
What evidence is there for 'accretion'? The current datasets include the Ha image and Ha velocity along the line of sight. Simulations have shown that the galaxy merger process can lead to tidal tails that drive gas out of the galaxy. I hope the authors could provide some comments about the accretion.
Author Response
The answer to the referee's concern about 'accretion flows' mentioned in the abstract and in the Conclusions.
The permanent outer cold gas accretion as a driver of disk galaxy evolution is now a commonly accepted paradigm. However, despite the success of the theory, observational findings of such accretion flows are still inconclusive. The aim of our paper is to search for possible outer accretion flows in panoramic spectral observations. And we have found them in several interacting systems. Yes, indeed, during interaction a galaxy gives off its gas into surrounding space; but all the dynamical simulations, e.g. by Hernquist with co-authors or e.g. by Mapelli (2008), demonstrate late re-accretion of the gas expelled. In a tight galaxy group, in particular, another galaxy may re-accrete this gas, which is different from the initial gas host. Just these possibilities are discussed in our paper as a possible source of outer gas accretion. Now I have added some remarks into the text (enhanced by boldface) which stress out this idea.
Reviewer 2 Report
Comments and Suggestions for Authors
The authors have addressed my main sources of concern.
Comments on the Quality of English Language
The English text still needs revision.
Author Response
Thank you for your estimate of our work over your previous comments.
As for the English, I have additionally edited the text, to improve the situation with articles (which are especially difficult for Russians) and some other grammar problems.